# [^123^I]-Meta-Iodobenzylguanidine Scintigraphy in Sarcoidosis: Exploring Cardiac Autonomic Dysfunction in Patients with Unexplained Cardiac Symptoms

**DOI:** 10.3390/diagnostics15182306

**Published:** 2025-09-11

**Authors:** Lisette R. M. Raasing, Marjolein Drent, Ruth G. M. Keijsers, Andor F. van den Hoven, Marco C. Post, Jan C. Grutters, Marcel Veltkamp

**Affiliations:** 1ILD Center of Excellence, Member of European Reference Network-Lung, St. Antonius Hospital, 3435 CM Nieuwegein, The Netherlands; m.drent@hetnet.nl (M.D.); m.veltkamp@antoniusziekenhuis.nl (M.V.); 2Department of Pharmacology and Toxicology, Faculty of Health, Medicine and Life Sciences, Maastricht University, 6200 MD Maastricht, The Netherlands; 3ILD Care Foundation Research Team, 6711 NR Ede, The Netherlands; 4Department of Nuclear Medicine, St. Antonius Hospital, 3435 CM Nieuwegein, The Netherlandsa.van.den.hoven@antoniusziekenhuis.nl (A.F.v.d.H.); 5Department of Cardiology, St. Antonius Hospital, 3435 CM Nieuwegein, The Netherlands; 6Division of Cardiology, University Medical Center Utrecht, 3584 CX Utrecht, The Netherlands; 7Division of Heart and Lungs, University Medical Center Utrecht, 3584 CX Utrecht, The Netherlands

**Keywords:** autonomic dysfunction, cardiac disease, MIBG, sarcoidosis, small fiber neuropathy

## Abstract

**Background/Objectives**: Sarcoidosis is a systemic inflammatory disease that can cause cardiac autonomic dysfunction (SCAD), often underrecognized despite its clinical importance. While [^18^F]fluoro-2-deoxy-D-glucose positron emission tomography/computed tomography ([^18^F]FDG PET/CT) and cardiac magnetic resonance imaging (CMR) assess cardiac involvement, [^123^I]-meta-iodinebenzylguanidine ([^123^I]MIBG) scintigraphy evaluates cardiac sympathetic innervation, offering complementary insights to potentially improve SCAD detection and management. This retrospective study explores the role of [^123^I] MIBG scintigraphy in detecting SCAD among patients with unexplained cardiac symptoms. It focuses on its potential to provide complementary diagnostic information in patients where established imaging techniques, such as [^18^F]FDG PET/CT and CMR, fail to detect cardiac sarcoidosis. **Methods**: Sarcoidosis patients referred to the St. Antonius Hospital (2017–2024) who underwent [^123^I]MIBG scintigraphy were included. Collected data encompassed demographics, SCAD symptoms, cardiac imaging findings, and carvedilol treatment outcomes. [^123^I] MIBG abnormalities were defined as a heart-to-mediastinal ratio ≤1.6 or a washout rate ≥20%. **Results**: Among the final cohort of 40 sarcoidosis patients with unexplained cardiac symptoms and normal [^18^F]FDG PET/CT and CMR findings, 19 patients (48%) showed abnormal [^123^I] MIBG scintigraphy results suggestive of SCAD. No significant differences were observed in clinical characteristics between patients with normal and abnormal [^123^I]MIBG findings. Of the 16 patients treated with carvedilol, 88% reported symptom improvement, although 50% experienced side effects. **Conclusions**: [^123^I]MIBG scintigraphy revealed abnormalities in a substantial number of sarcoidosis patients with unexplained cardiac symptoms despite normal [^18^F]FDG PET/CT and CMR. These findings indicate a potential role for [^123^I]MIBG in detecting SCAD, but prospective studies are needed to confirm their clinical significance.

## 1. Introduction

Sarcoidosis is a systemic inflammatory disorder characterized by noncaseating granulomas that can affect multiple organs, including the lungs, skin, eyes, and the heart [1]. Among its potentially life-threatening complications is cardiac sarcoidosis, caused by granulomatous infiltration of the myocardium. This can result in conduction abnormalities, ventricular arrhythmias, and left ventricular dysfunction, with symptoms ranging from dyspnea and dizziness to severe cardiovascular events [1,2]. Comprehensive diagnostic evaluation, including cardiac magnetic resonance imaging (CMR) and [^18^F]fluoro-2-deoxy-D-glucose-positron emission tomography/computed tomography ([^18^F]FDG PET/CT), is critical for assessing myocardial involvement [3].

However, symptoms caused by autonomic dysfunction of the heart may closely resemble those of cardiac sarcoidosis, and this sarcoidosis-associated cardiac autonomic dysfunction (SCAD) cannot be reliably detected through conventional imaging such as CMR or [^18^F]FDG PET/CT [2,4]. SCAD can lead to palpitations, syncope, dizziness, or exercise intolerance and may result in serious complications, including arrhythmias and sudden cardiac death [5,6]. Importantly, small-fiber neuropathy (SFN), which is increasingly recognized as a common and disabling complication of sarcoidosis, frequently includes autonomic dysfunction. SFN-related symptoms such as dizziness, palpitations, and gastrointestinal dysregulation [7,8,9] may therefore overlap with or aggravate the cardiac manifestations of sarcoidosis. As SFN is often underdiagnosed or misattributed to other causes, its contribution to the overall symptom burden may remain unrecognized, despite its substantial impact on quality of life and daily functioning [10,11,12].

[^123^I]-meta-iodobenzylguanidine ([^123^I]MIBG) scintigraphy provides unique insights into myocardial sympathetic innervation and may identify areas of impaired function even when structural changes are absent on CMR or [^18^F]FDG PET/CT [13,14]. Its integration with conventional imaging may thus support a more comprehensive evaluation and management of patients with unexplained cardiac symptoms.

Therefore, the aim of this retrospective study is to evaluate the value of [^123^I]MIBG scintigraphy in identifying SCAD in patients with unexplained cardiac symptoms. In particular, we aim to investigate whether [^123^I]MIBG scintigraphy provides complementary diagnostic information in cases where traditional imaging modalities, such as [^18^F]FDG PET/CT and CMR, do not show evidence of cardiac sarcoidosis.

## 2. Materials and Methods

### 2.1. Study Design and Subjects

This retrospective study included patients with sarcoidosis who were referred to the outpatient clinic of St. Antonius hospital, a tertiary referral center for sarcoidosis and interstitial lung diseases (ILDs) in the Netherlands, and underwent [^123^I]MIBG scintigraphy between October 2017 and February 2024. The sarcoidosis diagnosis was confirmed following international guidelines [1]. Patient characteristics in combination with results from [^18^F]FDG PET/CT, CMR, and [^123^I]MIBG scintigraphy were discussed in a multidisciplinary team (MDT) consisting of pulmonologist, cardiologists, and nuclear medicine specialists.

At our center, [^123^I]MIBG scintigraphy is performed in selected patients with sarcoidosis following discussion with the multidisciplinary team (MDT). Patients were referred for MIBG only if no clear explanation for their cardiac symptoms could be identified despite extensive prior diagnostic work-up. This included (1) patients with therapy-resistant cardiac sarcoidosis, as confirmed by the MDT based on clinical presentation and multimodal imaging, and (2) patients with extracardiac sarcoidosis presenting with cardiac symptoms (e.g., palpitations, chest pain, or syncope) in whom cardiac involvement was not demonstrated by [^18^F]FDG PET/CT or CMR and in whom other structural heart disease was excluded.

Patients with therapy-resistant cardiac sarcoidosis were defined as those with persistent or progressive cardiac involvement despite standard immunosuppressive therapy. Clinically relevant cardiac symptoms, such as unexplained palpitations or syncope, were considered indicative of cardiac involvement if they could not be attributed to other causes after thorough evaluation. To minimize interpretation bias and ensure that any abnormalities detected on [^123^I]MIBG scintigraphy could not be attributed to other known cardiac pathologies, patients with abnormal findings on [^18^F]FDG PET/CT or CMR (*n* = 6) were excluded. Abnormal findings included markedly increased myocardial uptake on [^18^F]FDG PET/CT and focal wall motion abnormalities or pathological contrast enhancement on CMR. None of the included patients had Parkinson’s disease [13]. A negative [^123^I]MIBG scintigraphy result was considered sufficient to rule out SCAD. Patients with positive [^123^I]MIBG scintigraphy were regarded as having a SCAD-compatible pattern and were subsequently treated with carvedilol added to their existing therapy.

The study was approved by the local Ethics Committee (Medical Research Ethics Committees United, Nieuwegein, the Netherlands; R19.080, 13 November 2012), and participants provided written informed consent. The study adhered to the Declaration of Helsinki and Good Clinical Practice (GCP) guidelines.

### 2.2. Measurements

#### 2.2.1. Diagnosis of Cardiac Sarcoidosis

A cardiac sarcoidosis diagnosis was made according the Heart Rhythm Society (HRS) expert consensus criteria [3]. Both definite and probable cardiac sarcoidosis were classified as cardiac sarcoidosis, mainly as both outcomes are comparable [15].

#### 2.2.2. Diagnosis of Cardiac Autonomic Dysfunction

[^123^I]MIBG scintigraphy was performed in accordance with the EANM guidelines [16], thereby ensuring methodological standardization and reproducibility. Planar anterior images were acquired 15 min and 4 h after tracer injection, with an acquisition time of 10 min per image using a low-energy high-resolution (LEHR) collimator and a matrix size of 128 × 128. The energy window was centered on the 159 keV photon peak with a 20% window. For region of interest (ROI) analysis, the cardiac ROI was manually delineated on the 15 min anterior image based on the myocardial contour and subsequently copied to the 4 h image, with adjustments applied as required. A rectangular ROI was drawn over the upper mediastinum on the 15 min image and transferred to the 4 h image. Although intra- and inter-observer reproducibility analyses were not performed in the present study, previously published data using this approach have demonstrated excellent reproducibility, with less than 5% intra- and inter-observer variability [17].

The [^123^I]MIBG tracer, a norepinephrine analog, is absorbed by cardiac sympathetic nerves and accumulates in presynaptic vesicles without further metabolism [18]. This allows for an evaluation of myocardial sympathetic innervation through subsequent scintigraphy imaging (Figure 1). Patients with cardiac autonomic dysfunction exhibit reduced uptake or decreased washout rates due to the impaired function of small nerve fibers. Abnormal results were defined as a heart-to-mediastinal uptake ratio (H/M) ≤ 1.6 and/or myocardial washout rate ≥ 20% after 4 h. The H/M ratio was calculated by placing ROIs over the heart and mediastinum on planar images (Figure 1B–E), with the ratio determined as mean counts in the heart ROI divided by mean counts in the mediastinum ROI. The myocardial washout rate was calculated by comparing counts in the heart ROI between early (15 min, Figure 1B,D) and delayed (4 h, Figure 1C,E) images, corrected for radioactive decay. Scintigraphy demonstrated diminished myocardial uptake or increased washout, indicating altered norepinephrine reuptake or release. Patients with decreased [^123^I]MIBG uptake were classified as having SCAD [19].

#### 2.2.3. Fluordeoxyglucose Positron Emission Tomography/Computed Tomography ([^18^F]FDG PET/CT)

[^18^F]FDG PET/CT examination was performed with a TF-64 combined PET/CT device (Philips Gemini, Medical systems, Eindhoven, The Netherlands), as described previously [20]. Patients were instructed to have a carbohydrate-restricted diet for 24 h followed by a fast of at least 6 h before injecting [^18^F]FDG. Dosage was based on body weight. An amount of 50 IEH/kg unfractionated heparin was pre-administered intravenously to suppress physiologic uptake in the myocardium, with a maximum of 5000 IE.

#### 2.2.4. Cardiac Magnetic Resonance Imaging

CMR images were acquired using a 1.5 T Philips MRI scanner with an eight-element phase array cardiac coil. A vector electrocardiographic system was used for cardiac gating. A stack of short-axis cine slices of both the right and left ventricle (8 mm thickness, no gap) from the base to the apex of the entire heart were acquired. If performed, T2-weighted short-tau inversion recovery images (indication myocardial edema) with 8 mm slice thickness were acquired at short-axis orientation. Late gadolinium enhancement (LGE) images were obtained 12–20 min after an intravenous administration of 0.4 mL/kg gadolinium [20].

### 2.3. Clinical Data

Patient characteristics, clinical diagnoses, and the outcomes of carvedilol treatment were retrospectively analyzed. Data on age, sex, body mass index (BMI), and sarcoidosis-related factors (e.g., treatment history, presence of cardiac sarcoidosis, and [^18^F]FDG PET/CT findings) were collected. The presence of SCAD symptoms and the evidence of SFN, based on documented clinical diagnoses, were included. In addition, patients were evaluated for other potential contributors to persistent symptoms, including anxiety disorders, metabolic abnormalities, and medication effects.

Information on the initiation of carvedilol or alternative treatments following abnormal [^123^I]MIBG scintigraphy was gathered, including therapy duration and any observed effects or adverse events. Therapy outcomes were assessed as patient-reported outcome measures and classified into three categories, namely “improved”, “no response”, or “worsened” based on clinical observations documented in the electronic patient records.

### 2.4. Statistical Analysis

To compare patients with and without SCAD, statistical analyses were performed with SPSS version 28.0 software for Windows (SPSS Inc., Chicago, IL, USA). Differences in mean age and BMI were assessed using Student’s *t*-test. The Mann–Whitney *U* test was used to compare time since sarcoidosis diagnosis. Associations between sex, cardiac sarcoidosis, SFN, use of medication, and symptoms of SCAD were tested using the chi-squared test. A *p*-value < 0.05 was considered significant.

## 3. Results

Preliminary results from this study were previously reported in a chapter of the first author’s PhD thesis [21]. In the present manuscript, we provide a more comprehensive and extended analysis of these findings. Between October 2017 and February 2024, 1214 patients with suspected cardiac sarcoidosis were evaluated by the cardiac sarcoidosis MDT. Indications for MDT evaluation were mainly the presence of symptoms suggesting cardiac sarcoidosis, including dizziness and palpitations, or incidental findings in the myocardium on [^18^F]FDG PET/CT made to evaluate extracardiac sarcoidosis. Of the 1214 patients, 59 (5%) underwent [^123^I]MIBG scintigraphy because of either therapy-resistant cardiac sarcoidosis or convincing cardiac symptoms that could not be explained by other cardiac conditions. Of these, 46 patients provided informed consent for the use of their data in this study. Among these 46 symptomatic sarcoidosis patients, 40 had normal findings on both [^18^F]FDG PET/CT and CMR. Demographic and clinical characteristics are summarized in Table 1. No significant differences were observed in demographic characteristics or [^18^F]FDG PET/CT findings between patients with and without abnormal [^123^I]MIBG scintigraphy (Table 1). Among these 40 patients, 19 (48%) showed abnormalities suggestive of SCAD, while the remaining 21 (52%) had normal scans. Although some had comorbidities potentially contributing to their symptoms, most still displayed features of autonomic dysfunction. The early heart-to-mediastinum (H/M) ratio at 15 min did not differ significantly between patients with normal and abnormal scans (*p* = 0.09). In contrast, the delayed H/M ratio at 4 h was significantly lower (*p* = 0.03) and the washout rate significantly higher (*p* < 0.0001) in patients with abnormal scintigraphy results (Figure 2).

No differences in presenting symptoms were observed between patients with abnormal [^123^I]MIBG scans and those with normal scans (Table 2).

Eighty-four percent (*n* = 16) of patients with an abnormal [^123^I]MIBG scintigraphy suggestive of SCAD started therapy with carvedilol as a result of this outcome. Assessed patient-reported outcome measures showed that 14 of these patients (88%) experienced a subjective improvement in their cardiac symptoms. However, 50% of patients discontinued carvedilol due to side effects, which included dizziness, fatigue, nausea, dry mouth, headache, and muscle ache. A comparison between patients who experienced side effects (*n* = 7) and those who did not (*n* = 8) revealed no significant differences in age, sex, BMI, time since diagnosis, or presence of cardiac sarcoidosis. As detailed in Table 3, most patients (*n* = 8) were treated with the lowest carvedilol dose (3.125 mg twice daily), while a smaller number received higher doses. Table 3 further summarizes the applied dose adjustment strategies and alternative treatment attempts, showing that four patients switched to another therapy after carvedilol intolerance and indicating whether these alternatives were associated with side effects. In nine patients, therapy duration was available, with a mean of 11 ± 16 months.

## 4. Discussion

In this retrospective study, we explored the potential role of [^123^I]MIBG scintigraphy in patients with sarcoidosis presenting with symptoms suggestive of cardiac involvement. [^123^I]MIBG scintigraphy abnormalities were detected in almost half of the sarcoidosis patients with unexplained cardiac symptoms, despite normal [^18^F]FDG PET/CT and CMR. These disturbances of SCAD may be underrecognized in sarcoidosis. However, in the absence of standardized diagnostic criteria for SCAD, this interpretation should be considered with caution.

The observed [^123^I]MIBG abnormalities may be explained by the underlying SFN, which is highly prevalent in sarcoidosis. SFN affects thinly myelinated and unmyelinated fibers and is strongly associated with autonomic dysfunction. The involvement of cardiac sympathetic nerve terminals may lead to impaired norepinephrine uptake and storage, detectable on [^123^I]MIBG scintigraphy even before structural myocardial damage becomes apparent. Thus, [^123^I]MIBG abnormalities may reflect early or subclinical sympathetic denervation, linking autonomic symptoms with functional rather than structural pathology. This interpretation aligns with previous reports associating SFN with cardiac sympathetic dysfunction in sarcoidosis [22]. Nevertheless, a normal [^123^I]MIBG scan does not exclude autonomic dysfunction, and persistent symptoms may reflect false negative findings or alternative etiologies. Future studies integrating [^123^I]MIBG scintigraphy with direct SFN assessments (e.g., skin biopsy or autonomic testing) are warranted to clarify this relationship and its therapeutic implications. Current guidelines highlight the significance of using [^18^F]FDG PET/CT and CMR in diagnosing myocardial involvement in sarcoidosis [2], but they do not address the management of unexplained functional symptoms. [^123^I]MIBG scintigraphy may provide additional insights by evaluating cardiac sympathetic innervation. It was previously demonstrated that [^123^I]MIBG scintigraphy can detect impaired cardiac sympathetic innervation in a significant subset of sarcoidosis patients, particularly in those with SFN, even when standard cardiac assessments show no abnormalities [22]. The presence of [^123^I]MIBG abnormalities in patients with significant cardiac symptoms but unremarkable findings on standard imaging raises important questions about the prevalence and clinical relevance of SCAD. This highlights the need for further studies to better define its diagnostic utility and therapeutic implications. However, while [^123^I]MIBG scintigraphy may provide additional insights in this population and can detect sympathetic dysregulation, it should be noted that SCAD remains a clinical construct lacking standardized diagnostic criteria. This underscores the need for consensus definitions and standardized testing protocols in future research.

Although recent cardiac sarcoidosis guidelines emphasize [^18^F]FDG PET/CT and CMR for diagnosing myocardial involvement, they also recognize the importance of addressing unexplained functional symptoms. SCAD may contribute to cardiac arrhythmias, while granulomatous inflammation within the heart can also directly lead to cardiac arrhythmias. In this context, [^123^I]MIBG scintigraphy’s potential utility lies in bridging the gap when autonomic dysfunction is suspected [23,24]. Cardiac sympathetic activity is vital for heart function, and [^123^I]MIBG scintigraphy non-invasively evaluates sympathetic terminal integrity, detecting dysfunction associated with worse prognosis in various cardiac diseases [23]. [^123^I]MIBG scintigraphy has demonstrated the capability to detect disruptions in the adrenergic nervous system among patients with sarcoidosis, often exhibiting greater sensitivity compared to thallium scintigraphy [25]. Additionally, reduced [^123^I]MIBG uptake was identified as a predictor of fatal arrhythmias, highlighting its potential utility in risk stratification for interventions such as implantable cardioverter defibrillator (ICD) therapy [26].

Treatment with carvedilol demonstrated beneficial effects in SCAD, particularly in heart failure and autonomic disorders, by mitigating sympathetic overactivity [27]. Clinically, case reports have described sarcoidosis patients with severe SCAD who showed improvement after carvedilol treatment, supporting its therapeutic role [27,28]. The present study offers a preliminary but promising indication of carvedilol’s effectiveness in alleviating the symptoms of SCAD among patients exhibiting abnormalities on [^123^I]MIBG scintigraphy. Of the cohort, 80% commenced carvedilol treatment, with 88% reporting symptomatic improvement, although 50% experienced adverse effects such as dizziness, fatigue, and nausea.

However, the predictors of carvedilol tolerability were not formally analyzed due to the small number of patients who discontinued therapy. Carvedilol is primarily metabolized in the liver via CYP2D6 and CYP2C9, highlighting an important area for future investigation [29]. These findings should be considered preliminary given the retrospective design and small sample size.

This study has several limitations. The retrospective design and small sample size limit causal inference, introduce potential selection bias, and reduce generalizability. Patients were selected based on MDT judgement rather than consecutive inclusion, which may have introduced additional referral bias. The presence of [^123^I]MIBG abnormalities in nearly half of our patients despite normal findings on conventional imaging may be explained by underlying SFN. Other validated autonomic assessments, such as heart rate variability analysis, Ewing battery, Holter monitoring, and tilt table testing, were not utilized, further constraining the depth of evaluation [30]. [^123^I]MIBG is primarily considered a second-line diagnostic modality, typically reserved for patients with unexplained cardiac symptoms, syncope, or small fiber neuropathy, which may limit the generalizability of these observations. Technical factors, including limited spatial resolution and the absence of standardized acquisition and interpretation protocols, may also compromise sensitivity and reproducibility. In addition, SCAD remains a clinically defined entity without universally accepted diagnostic criteria, complicating the interpretation of scintigraphic findings. Finally, the limited availability of [^123^I]MIBG, together with associated cost and logistical challenges, constrains broader clinical implementation. Prospective, multicenter studies with standardized protocols and a comprehensive range of autonomic assessments are warranted to confirm and expand these results [23].

## 5. Conclusions

Our findings suggest that [^123^I]MIBG scintigraphy may offer additional value in detecting functional disturbances compatible with SCAD, particularly in patients with pronounced clinical symptoms and inconclusive findings on conventional cardiac imaging. In our cohort, this applied to almost half of the patients. These results underscore the need for further prospective studies to clarify the clinical utility of [^123^I]MIBG scintigraphy and to explore its potential role in guiding personalized treatment strategies. Given the absence of standardized diagnostic criteria for SCAD, these interpretations should be considered with caution. Until more robust evidence becomes available, the use of [^123^I]MIBG scintigraphy should be reserved for selected patients within specialized centers.

## Figures and Tables

**Figure 1 diagnostics-15-02306-f001:**
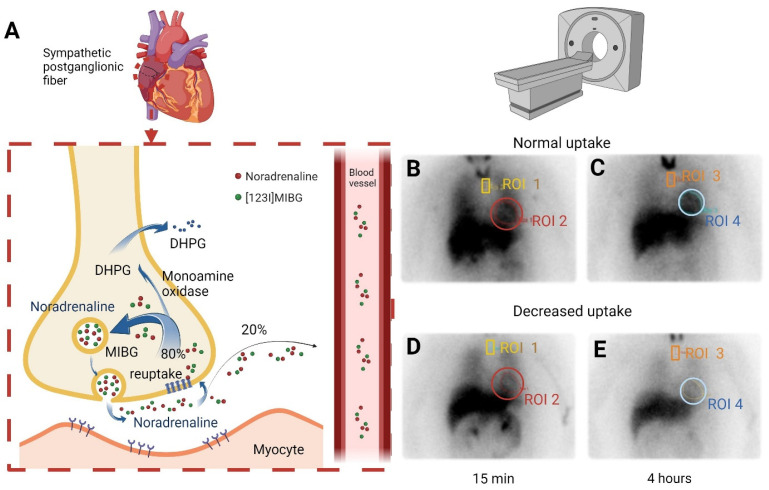
(**A**) Illustration of the mechanism of [^123^I]MIBG scintigraphy. A norepinephrine analog is absorbed by cardiac sympathetic nerves and accumulates in presynaptic vesicles without further metabolism. (**B**) Normal uptake at baseline; ROI 1 = mediastinum, ROI 2 = heart, heart-to-mediastinal uptake ratio = ROI 2/ROI 1. (**C**) Normal uptake after 4 h; ROI 3 = mediastinum, ROI 4 = heart, heart-to-mediastinal uptake ratio = ROI 4/ROI 3. (**D**) Decreased uptake at baseline; ROI 1 = mediastinum, ROI 2 = heart, heart-to-mediastinal uptake ratio = ROI 2/ROI 1. (**E**) Decreased uptake after 4 h; ROI 3 = mediastinum, ROI 4 = heart, heart-to-mediastinal uptake ratio = ROI 4/ROI 3. DHPG: dihydroxyphenylglycine; [^123^I]MIBG: 123I-metaiodobenzylguanidine; ROI: region of interest.

**Figure 2 diagnostics-15-02306-f002:**
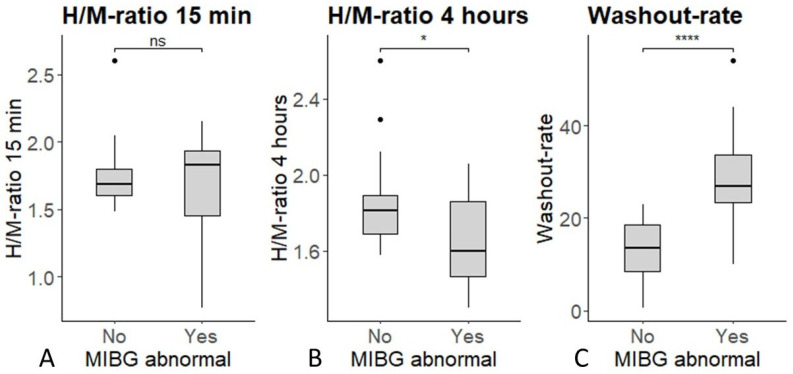
Boxplots comparing [^123^I]MIBG-derived parameters between patients with normal and abnormal scans: (**A**) early H/M ratio (15 min), (**B**) delayed H/M ratio (4 h), and (**C**) washout rate. Statistical significance was defined as *p* < 0.05 (*), and *p* < 0.0001 (****).

**Table 1 diagnostics-15-02306-t001:** Summary demographic data and clinical data of patients with sarcoidosis having normal findings on both fluordeoxyglucose positron emission tomography and cardiac magnetic resonance imaging who underwent an [^123^I]-meta-iodobenzylguanidine ([^123^I]MIBG) scintigraphy, subdivided in normal and abnormal [^123^I]MIBG scintigraphy.

	Total Group (*n* = 40)	[^123^I]MIBG Normal (*n* = 21)	[^123^I]MIBG Abnormal (*n* = 19)	*p*-Value
Age, years, mean ± SD	48 ± 11	45 ± 9	52 ± 12	NS
Male, *n* (%)	23 (58)	12 (57)	11 (58)	NS
BMI kg/m^2^, mean ± SD	28 ± 5	28 ± 4	28 ± 5	NS
Time since diagnosis, years, median ± IQR	11 ± 7	11 ± 6	12 ± 12	NS
Cardiac sarcoidosis, *n* (%)	9 (23)	3 (14)	6 (32)	NS
Therapy % (*n*)				
No immunosuppressive therapy, *n* (%)	24 (60)	14 (67)	10 (53)	NS
Prednisone, *n* (%)	7 (18)	4 (19)	3 (16)	NS
Methotrexate, *n* (%)	8 (20)	3 (14)	5 (26)	NS
TNF-alpha inhibitors, infliximab or adalimumab, *n* (%)	2 (5)	1 (5)	1 (5)	NS
Methylprednisone, *n* (%)	1 (3)	1 (5)	0 (0)	NS
Adalimumab, *n* (%)	1 (3)	0 (0)	1 (5)	NS

*n*: number; SD: standard deviation; BMI: body mass index; IQR: interquartile range; TNF: tumor necrosis factor. Comparing [^123^I]MIBG scintigraphy abnormal vs. normal, all *p* values were > 0.05 (NS: not significant).

**Table 2 diagnostics-15-02306-t002:** Symptoms of patients with sarcoidosis analyzed for cardiac autonomic dysfunction subdivided in normal and abnormal [^123^I]-meta-iodinebenzylguadine ([^123^I]MIBG) scintigraphy.

	Total Group (*n* = 46)	[^123^I]MIBG Normal (*n* = 26)	[^123^I]MIBG Abnormal (*n* = 20)
Chest pain, *n* (%)	14 (30)	7 (27)	7 (35)
Arrhythmia, *n* (%)	8 (17)	6 (23)	2 (10)
Collapse, *n* (%)	4 (9)	1 (4)	3 (15)
Near collapse, *n* (%)	3 (7)	2 (8)	1 (5)
Syncope, *n* (%)	1 (2)	1 (4)	0 (0)
Orthostasis, *n* (%)	1 (2)	1 (4)	0 (0)

Data are expressed as percentage with total number in parentheses. Comparing abnormal vs. normal [^123^I]MIBG scans, all *p* values were >0.05 (not significant).

**Table 3 diagnostics-15-02306-t003:** Carvedilol dosing strategies, treatment discontinuation due to side effects, and subsequent alternative therapy attempts with reported adverse effects (*n* = 16).

Dose (Twice Daily)	*n*	Alternative Therapy Strategy	Adverse Effects?	*n*
3.125 mg	8	Nebivolol	Yes	1
3.125–6.25–3.125 mg	1	Valsartran	Yes	1
3.125–6.25–12.5 mg	1	Promocard	No	1
6.25–12.5–25 mg	1	Amlodipine	Unknown	1
12.5 mg	2	Unknown		2
25 mg	1	None		10
Unknown	2			

## Data Availability

The data presented in this study are available on request from the corresponding author. The data are not publicly available due to privacy.

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
