# Peer review of "[123I]-Meta-Iodobenzylguanidine Scintigraphy in Sarcoidosis: Exploring Cardiac Autonomic Dysfunction in Patients with Unexplained Cardiac Symptoms"

_diagnostics, 2025, doi:10.3390/diagnostics15182306_

Round 1
Reviewer 1 Report
Comments and Suggestions for Authors
This study focuses on sarcoidosis patients experiencing unexplained cardiac symptoms despite negative results from conventional imaging methods (FDG PET/CT and CMR). It utilizes 123I-MIBG assess cardiac sympathetic nervous function. The results indicate abnormal uptake activity in 48% of patients, with most of whom showed symptomatic improvement after carvedilol treatment. However, due to its retrospective design, small sample size, and lack of autonomic function reference tests, the conclusions should be interpreted with caution. The study demonstrates innovation and provides suggestive insights, but the current level of evidence is limited.
- Methods & Experimental Design
- Lack of MIBG Technical Details: The following core parameters were not explicitly described: Image acquisition time, collimator type, Energy window settings, region of Interest (ROI) delineation method, and intra- and inter-observer reproducibility validation data. Please provide the above details to enhance methodological reproducibility.
- Lack of Pre-/Post-Therapy Imaging Comparison: The authors suggest carvedilol improves symptoms by modulating autonomic balance, but rely solely on subjective patient reports (88% symptom improvement) without objective assessment. If feasible, please provide the comparison of pre- and post-treatment MIBG parameters (e.g., changes in H/M ratio, wash-out rate) or other parameters which could objectively quantify sympathetic nervous function improvement.
- Insufficient Analysis of Carvedilol Side Effects: Treatment was discontinued due to side effects in 50% of patients, but the dose adjustment strategy or alternative treatment options were not detailed.
- Results Presentation
- Insufficient Data Visualization: Table 1 only provides basic demographic characteristics. Please add distribution plots or boxplots of MIBG semi-quantitative results (H/M values, wash-out rates) to visually illustrate differences between the normal and abnormal uptake groups.
- Suggestions for Enhancing Discussion Depth
- Weak Mechanistic Explanation: The pathological basis for MIBG abnormalities in 48% of patients despite negative conventional imaging is not deeply explored. Please link findings to small fiber neuropathy (SFN): Could this be due to sympathetic nerve terminal damage preceding structural myocardial changes? Deepen the discussion by incorporating the pathological link between SFN and autonomic dysfunction.
- MIBG's Clinical Role Unclear: MIBG was not compared to other autonomic assessment tools (e.g., cold pressor test) regarding sensitivity. Should MIBG be positioned as a second-line screening tool? Which symptom combinations (e.g., syncope + SFN) hold the highest predictive value?
- Neglect of False-Negative Population: Symptoms persisted in the 52% of patients with normal MIBG. Analyze other potential etiologies (e.g., anxiety disorders, metabolic diseases) to avoid over-attribution to SCAD.
- Formatting & Terminology
- Terminology Consistency: The term "Washout rate" appears inconsistently as both "washout" and "wash-out". Please standardize usage throughout the whole text.
- Reference Number Issue: The first sentence of the introduction cites references [8], but preceding numbers [1]-[7] are missing. Please verify the sequential continuity of reference number throughout the whole manuscript.
None
Reviewer 2 Report
Comments and Suggestions for Authors
I have appropriately reviewed the case report entitled “Meta-iodobenzylguanidine scintigraphy in sarcoidosis: exploring cardiac autonomic dysfunction in patients with unexplained cardiac symptoms”. The manuscript addressed an understudied clinical problem: the role of [¹²³I]MIBG scintigraphy in detecting sarcoidosis-associated cardiac autonomic dysfunction (SCAD) in patients with unexplained cardiac symptoms and normal FDG PET/CT and CMR findings. The study highlights an important diagnostic gap in cardiac sarcoidosis evaluation. The finding that nearly half of symptomatic patients with normal PET/CT and CMR had abnormal MIBG results is clinically meaningful. However, the interpretation of these results as indicative of SCAD needs caution, given the lack of standardized diagnostic criteria for SCAD. The authors should emphasize this limitation in both the Results and Discussion sections. The retrospective design of the study is a significant limitation, as acknowledged. The inclusion and exclusion criteria are clearly stated, but more details about patient selection are needed. For example: Were patients consecutively included, or was there a potential referral bias for MIBG scintigraphy? Please include the symptoms of the patients accordingly. How were “therapy-resistant cardiac sarcoidosis” and “convincing cardiac symptoms” operationally defined? SCAD is defined solely on abnormal [¹²³I]MIBG findings plus symptoms. Since autonomic testing (e.g., tilt-table, HRV analysis) was not used, the authors should clarify that MIBG abnormalities may reflect—but do not confirm—SCAD. This distinction is critical for avoiding overinterpretation. The reported symptomatic improvement in 88% of treated patients is promising, but the high discontinuation rate (50%) due to side effects is equally important. The authors should provide more detail on dose ranges, treatment duration, and management of adverse effects, as this will guide clinical applicability. Additionally, it would be useful to compare patients who tolerated carvedilol versus those who discontinued, to explore whether baseline characteristics predicted tolerability. Please soften your conclusion to reflect that findings are preliminary and require prospective validation in the abstract section. Please consider shortening and more focusing on the main hypothesis in the Introduction section. The section “Diagnosis of cardiac sarcoidosis” is duplicated (2.2.1 and 2.2.2 headings). Please correct accordingly. The authors should also emphasize the practical barriers of MIBG implementation (availability, standardization, cost).
Reviewer 3 Report
Comments and Suggestions for Authors
Dear Authors,
this is an interesting paper that focuses on a new field of research. Some issues are however present:
- [18F]FDG and [123I]MIBG need to be properly written all the times that they are cited;
- line 128: some examples of abnormal findings will help to strengthen the value of the paper;
- it is not clear if patients with known cardiological pathologies, which can result in reduction of both [18F]FDG and [123I]MIBG have been excluded. Similarly, patients with Parkinson’s disease need to be excluded for the same reason;
- paragraphs 2.2.1 and 2.2.2 have the same title;
- in paragraph 2.2.2 it should be explained how H/M ratio and washout were calculated, aiding therefore the comprehension of Figure 1;
- in Table 1 and in the results no p-value have been reported, therefore it is not possible to have a clear comprehension of the meaning of the whole paper;
- semiquantification through H/M ratio and washout have been calculated, however there are not presented in the result section and no statistical analysis have been performed on these value;
- clearly the lack of a precise and “quantitative” evaluation of the response to carvedilol through specific examination (e.g. tilt test, Ewing battery, holter, ..) is missing. This is a major issue of the paper that needs to be addressed in some ways;
Best regards.
Round 2
Reviewer 2 Report
Comments and Suggestions for Authors
The authors actually replied to all my previous comments. I don't have any more comments.
Reviewer 3 Report
Comments and Suggestions for Authors
All my comments have been addressed, enhancing the overall quality of the paper.